# Physiological and Nutritional Responses of Ungrafted Merlot and Cabernet Sauvignon Vines or Grafted to 101-14 Mgt and 1103P Rootstocks Exposed to an Excess of Boron

Kleopatra-Eleni Nikolaou [1,*], Theocharis Chatzistathis [2] , Serafeim Theocharis [1] , Anagnostis Argiriou [3] and Stefanos Koundouras [1]

[1] School of Agriculture, Aristotle University of Thessaloniki, 54124 Thessaloniki, Greece
[2] Hellenic Agricultural Organization (H.A.O.) 'Demeter', Institute of Soil and Water Resources, 57001 Thessaloniki, Greece
[3] Institute of Applied Bioscience, 6th Km Charilaou–Thermi Road, 60361 Thessaloniki, Greece
[*] Correspondence: nikolaouk@agro.auth.gr

**Abstract:** The aim of this study is to analyze the effect of excess boron (B) on the nutrient uptake, growth, and physiological performance of grapevines. Merlot and Cabernet Franc grapevines, either own-rooted or grafted onto 1103P and 101-14 Mgt rootstocks, were exposed in a hydroponic pot experiment to 0.5 mM boron for sixty days. Twenty-five days following the beginning of B treatment, the first symptoms of boron toxicity appeared, including leaf edge and margin yellowing, subsequent necrosis, and cupping of leaf blades. At harvest, sixty days after the start of the experiment, B concentration of the treated vines increased in all parts of the vine in the following order: Leaves > Roots > Trunks > Shoots. Leaf Boron concentration in treated vines ranged from 980.67 to 1064.37 mg kg$^{-1}$ d.w. Boron excess significantly reduced the concentrations of all macro and micronutrients studied in this experiment. The total leaf chlorophyll (Chl) concentration decreased from 35.46 to 44.45%, thirty and sixty days, respectively, from the beginning of the boron treatments. In addition, an excess amount of boron resulted in a dramatic decrease in net $CO_2$ assimilation rate, stem water potential, and PSII maximum quantum yield, irrespective of vine type (own-rooted or grafted). At the end of the experimental period, the total leaf phenolic content increased by 71.73% in Merlot and by 71.16% in Cabernet Franc due to Boron stress. The tressed vines grafted onto 1103P showed increased shoot and root dry weights, leaf chlorophyll content, $CO_2$ assimilation rates, and $F_v/F_m$ ratio compared to vines grafted onto 101-14 Mgt. It was found that rootstocks play an important role in B toxicity. The results showed that the leaf accumulation of boron was delayed to a greater extent by 1103P rootstock compared to 101-14 Mgt, resulting in the earlier appearance of leaf toxicity symptoms in vines grafted onto 101-14 Mgt rootstock compared to 1103P.

**Keywords:** grapevine; boron toxicity; rootstocks; ion concentration; $CO_2$ assimilation; stem water potential; chlorophyll fluorescence



## 1. Introduction

Boron (B) is known as an essential element for plants, with a narrow range between its phytotoxic limit and deficient level in the soil [1,2]. It plays an important role in various physiological, cellular, and metabolic processes, including sugar transport, photosynthesis, cell differentiation, cell wall synthesis, and the generative growth of plants [3,4]. B is also required for nucleic acid synthesis and reproductive structure development [5]. Previous studies have reported that one of the primary functions of B concerns the structure of primary plant cell walls, connecting the glycosyl-inositol-phospho-ceramides (GIPCs) of the plasma membrane with the arabinogalactan proteins (AGPs) of the cell wall, thereby attaching the membrane to the cell wall [6,7], affecting the mechanical properties of the cell wall [8].

Boron is found in nature, most frequently combined with oxygen, in the form of borates, such as borax $Na_2B_4O_7 \bullet 10H_2O$. B appears less frequently as boric acid [$B(OH)_3$ or $H_3BO_3$] and rarely as $BF_4$ anion associated with fluorine [9]. Boron is released into the soil solution as boric acid, which can be readily taken by the plants or leached from the soil. Among the trace elements, B is often deficient due to its high mobility. In nature, there are few B-rich soils compared to B-deficient soils. The range between B deficiency and toxicity is relatively narrow, thus B often becomes toxic for plants even in slightly high concentrations [10]. In cases of boron deficiency in plants, boron must be carefully supplied [11]. In addition, high levels of B can be an important abiotic stress factor in the world, negatively affecting crop yield and plant development [12]. In conditions of B toxicity, B binds to the ribose moieties of biological molecules, causing disruptions in cell development and division, as well as alterations in cell wall structure [13]. Sensitivity to B toxicity apparently involves several metabolic processes [13,14], such as photosynthetic rate reduction and negative effects on photosystem II efficiency, reduction of photosynthate supply of developing parts, increased membrane permeability, as well as diminution of protons extrusion in the root system and nutritional imbalances.

High B concentrations may occur naturally in the soil or in groundwater or may be added to the soil from mining, fertilizers, or irrigation water [15]. The most important contributor among the different potential sources leading to high levels of B in the soil is irrigation water [16]. According to Kloppmann et al., 2005 [17], entire Mediterranean regions suffer from scarce groundwater and surface water boron (B) contamination, rendering them inadequate for irrigation purposes of some plant species. One of the widely used techniques to avoid B toxicity is the grafting technique. Previously published works have demonstrated that the use of different rootstocks can considerably influence the tolerance of scion to B toxicity [18,19]. However, nowadays, little is known about the response of different grapevine scion–rootstock combinations in conditions of Excess B.

Grapevine (*Vitis vinifera* L.) is one of the most commercially grown fruit crops worldwide. It is commonly grown when grafted onto different rootstocks. In ancient times, grafting was already a widespread practice; however, the main reason for the use of grafting in viticulture was the *Daktulosphaira vitifoliae* (phylloxera) epidemic. At the end of the nineteenth century, Phylloxera, which was native to North America at that time, was introduced into Europe and destroyed millions of vineyard hectares. To prevent severe damage to the root system due to phylloxera, rootstocks from the Vitis species originating from North America were employed. Apart from their ability to help scion cope with phylloxera, rootstocks can also confer tolerance to a wide range of abiotic stresses.

Little is known about the response of grapevines to relatively high boron levels. According to Maas in 1990 [20], grapevines have been identified as being sensitive to excess B. According to Peacock and Christensen [11], irrigation water with a B concentration higher than 0.1 mM may cause toxicity in grapes. It is also mentioned that 0.1 mM B in saturated soil extracts and 80 mg·kg$^{-1}$ B in leaf tissue is indicative of B toxicity. Growth reduction under B-toxic conditions was reported by Gunes et al. in 2006 [21]. Nikolaou et al., 1995 [22] reported severe toxicity symptoms in Victoria grapes when irrigated for five years with water containing 0.4 mM B.

In terms of Greece, extensive contamination of irrigation water with B was found in a viticultural region located about forty kilometers southeast of Thessaloniki, Northern Greece [22].

Thus, the objectives of this study were to investigate the effects of excess B on some physiological parameters, growth, boron and other nutrient element concentration and uptake in various organs of non-grafted and grafted vine plants. Specifically, two vine rootstocks were used (1103P and 101-14 Mgt) to assess their effects on vine tolerance to B toxicity.

## 2. Material and Methods

This study was carried out at the experimental farm of the Aristotle University of Thessaloniki, which is located 15 Km southeast of Thessaloniki (Northern Greece, geographical co-ordinates: N 40°32.267′ E 22°59.885′). Two-year-old vine plants of Merlot and Cabernet Franc cultivars, either own-rooted or grafted onto 1103P and 101-14 Mgt rootstocks, were planted at the beginning of the 2019 growing season. Pots with a capacity of 10 L were used to plant the vine plants, containing a 9 L medium of inert sand and perlite in a 1:1 ratio. After pruning the vine plants to single shoots with two buds, all pots were placed outdoors. For each scion–rootstock combination or own-rooted vine, six plants of similar stem diameter and height were selected, transported to the experimental location, and irrigated automatically every two days using a drip irrigation system with 650 mL per plant of a modified 50% Hoagland No. 2 nutrient solution [23].

The two vine rootstocks used in our experiment were chosen based on their genetic character: the 1103P rootstock, a hybrid of *V. berlandieri*/*V. rupestris*, is well adapted to deep, well-drained calcareous soils, offers excellent phylloxera resistance, while the *V. riparia*/*V. rupestris* hybrid, 101-14 Mgt rootstock, which tends to be easy to root and graft and offers excellent protection against phylloxera, is not well adapted in calcareous soils.

The experiment was conducted based on a ($3 \times 2 \times 2$) factorial combination of own-rooted vines or two vine rootstocks, two vine cultivars, two levels of B treatment (Control and 0.5 mM B), and six replications. The treatment started at the beginning of July and continued for sixty days. During this period, 650 mL of 0.5 mM B solution was applied, in the form of boric acid three times a week to induce boron toxicity.

During the experimental planning, similar preliminary experiments determined the appropriate concentration of boron (the vines were exposed to different B concentrations ranging from 0 to 1 mM).

During the summer months, the vines were covered with a green polyethylene net to prevent overheating of leaves and root systems. When performing measurements of physiological parameters, the net could be easily removed to allow maximum light exposure, and then reattached. During the vegetative period, stem water potential, leaf chlorophyll content, and photosynthetic parameters were measured at three different stages: (1 July, 1 August, and 1 September, respectively).

### 2.1. Chlorophyll Content

Throughout the experimentation period, destructive and non-destructive methods were used to evaluate leaf chlorophyll content. A non-destructive method was applied to estimate chlorophyll content. The chlorophyll index was recorded on three fully expanded leaves located on basal nodes of the shoots using a chlorophyll content meter (CM) CCM-200 (Opti-Sciences, Tyngsboro, MA, USA). The same leaves were selected to determine the chlorophyll content using Wintermans and De Mots', 1965 [24] method: Leaf discs of approximately 0.5 g were extracted in 15 mL of 96 percent ethanol before being placed in a water bath at 79.8 °C for about 2 h, or until completely discolored. A spectrophotometer (Jenway Ltd., Essex, UK) was employed to determine the absorbance of the extract at 665 nm for chlorophyll a, and 649 nm for chlorophyll b, respectively. The total chlorophyll was determined using the following equation.

$$\text{Cl (a + b) mg g}^{-1} \text{ FW} = (6.10 \times A665 + 20.04 \times A649) \times 15/100 \text{ FW} \tag{1}$$

### 2.2. Stem Water Potential and Gas Exchange

According to Chone et al., 2001 [25], the water status of vines was estimated by measuring stem water potential (SWP) at three stages during the experimentation period using a pressure chamber. Each selected leaf was firmly enclosed in a polythene bag to prevent transpiration. Aluminum foil was also placed around the polythene bag for at least 90 min prior to measurement. The Plant Efficiency Analyser (PEA Hansatech Instruments Ltd., King's Lynn, UK) was used to measure chlorophyll fluorescence in attached, 30 min

dark-adapted leaves between 11:00 a.m. and 1:00 p.m. Actinic light (635 nm) with a solid weak pulse of 3500 μ mols m$^{-2}$ s$^{-1}$ PPFD (Photosynthetic Photon Flux Density) was used for light exposure. By measuring the fluorescence signal from the dark-adapted leaves (Minimum $F_0$, Maximum $F_m$, and variable $F_v$ fluorescence yield parameters), the maximum quantum efficiency of photosystem II (PSII) was calculated as $F_v/F_m$ ($F_v = F_m - F_0$).

Leaf gas exchange was measured on 3 August 2019 and 3 September 2019, on healthy, intact, and well-developed leaves adjacent to those used for SWP, using a portable LCi gas exchange system (ADC BioScientific Ltd., Hoddesdon, UK). Based on gas exchange measurements, stomatal conductance (gs) and the net $CO_2$ assimilation rate (A) were estimated.

Measurements were taken under the following conditions: Intercellular $CO_2$ concentration (Ci) ranged from 124 to 311 μmol mol$^{-1}$ on 3 August 2019 and from 111 to 276 μmol mol$^{-1}$ on 3 September 2019. Leaf temperature (TI) ranged from 36.2 to 43.1 °C on 3 August 2019 and from 33.7 to 39.7 °C on 3 September 2019. The photon flux density was kept constant at 1014 and 1013 μmol m$^{-2}$ s$^{-1}$ on 3 August and 3 September, respectively.

### 2.3. Tissue Nutrient Concentrations and Growth Parameters

The vines were harvested in early September at the end of the experimental period, and divided into leaves, stems, trunks, and roots. By submerging the pots in water and gently separating the potting sand from the root balls, roots were relieved from the sand. Roots that broke away from the balls were retrieved from the washing water with a sieve. Three 5 L baths of distilled water were used to rinse the leaves after harvest. An estimation of the dry weight was achieved after all plant material had been dried to a constant weight at 70 °C. The dried material was then ground into a fine powder that could pass through a 30-mesh screen. The fine powder of each sample (0.5 g) was heated to 515 °C in a muffle furnace for five hours, then dissolved in 3 mL 6 NHCl and diluted with double-distilled water to a volume of 50 mL. ICP-OES (Perkin Elmer-Optical Emission Spectrometer, OPTIMA 2100 DV, Ontario, ON, Canada) was used to estimate the concentrations of P, K, Ca, Mg, Na, Fe, Mn, Zn, and Cu, while Kjeldahl [26] and azomethine-H [27] were applied to measure N and B, respectively.

### 2.4. Total Phenolics

Sixty days following the beginning of the B treatments, leaf samples were collected to determine total phenolic content using the Folin–Ciocalteu colorimetric method [28]. Segments of fresh leaf blade weighing 0.3 g were submerged in 80% methanol. Standard curves were developed using catechin. In terms of the leaf's fresh weight, the phenolic concentrations were expressed as mg g$^{-1}$ catechin equivalents (CE)(f.w.). Each extract was subjected to a triplicate analysis.

### 2.5. Statistical Analysis

The experimental data were analyzed using analysis of variance (ANOVA) in the SPSS Version 25 program and the Least Significant Difference method, with a significance level of $p < 0.05$ applied to compare the differences.

### 2.6. Climate Data

The B treatments were conducted for sixty days during the months of July and August 2019. Monthly climate data during experimentation were as presented in Table 1.

**Table 1.** Monthly climate data, during the experimental period.

| Monthly Climate Data during Treatments | | | | | |
|---|---|---|---|---|---|
| Month | Tmean | Tmin | Tmax | Precipitation | Sunshine |
| July | 28.2 | 21.1 | 32.1 | 48 | 355 |
| August | 29.1 | 21.8 | 33.4 | 0 | 357 |

Tmean: mean temperature (°C), Tmin: minimum temperature (°C), Tmax: maximum temperature (°C), Precipitation (mm), Sunshine (hours).

## 3. Results

### 3.1. Boron Distribution in Different Vine Parts

Boron concentration in the root system, trunk and leaves of grafted and ungrafted vine plants are presented in Table 2. At harvest, sixty days from the beginning of the experimentation, 0.5 mM boron treatment significantly increased B levels of all vine parts, following the order: Shoots < Trunks < Roots < Leaves.

**Table 2.** The effect of boron toxicity on leaf, shoot, trunk, and root on B concentrations (mg kg$^{-1}$ d.w.) of own-rooted or grafted onto 1103P and 101-14 Mgt rootstocks of Merlot and Cabernet Franc vine cultivars.

| Treatments | | Leaves | | Shoots | | Trunks | | Roots | |
|---|---|---|---|---|---|---|---|---|---|
| | | M | CF | M | CF | M | CF | M | CF |
| Control | OR | 47.56 [d] | 56.89 [d] | 15.81 [c] | 13.53 [c] | 11.13 [b] | 11.45 [b] | 33.09 [c] | 34.88 [c] |
| | 1103P | 62.93 [d] | 59.76 [d] | 14.32 [c] | 13.09 [d] | 9.97 [b] | 10.71 [b] | 22.62 [d] | 34.17 [c] |
| | 101-14 Mgt | 53.15 [d] | 54.17 [d] | 15.24 [c] | 12.97 [d] | 10.46 [b] | 10.03 [b] | 35.23 [c] | 30.79 [c] |
| 0.5 mM B | OR | 1021.30 [b] | 1064.37 [a] | 20.69 [b] | 16.40 [c] | 23.51 [a] | 18.49 [a] | 52.84 [b] | 59.70 [a] |
| | 1103P | 980.67 [c] | 990.31 [c] | 26.32 [a] | 19.82 [b] | 27.46 [a] | 27.87 [a] | 52.21 [b] | 51.87 [b] |
| | 101-14 Mgt | 1011.70 [b] | 1033.18 [b] | 21.97 [b] | 17.87 [c] | 23.88 [a] | 19.67 [a] | 55.71 [a] | 62.72 [a] |
| LSD: $p < 0.05$ | | 19.60 | | 3.529 | | 11.06 | | 3.17 | |
| F < 0.001 | | 634.86 | | 9.399 | | 24.42 | | 45.008 | |

OR: Own root, M: Merlot, CF: Cabernet Franc. Different letters in each column represent significant differences at $p < 0.05$.

In treated vines, the B concentrations (mg kg$^{-1}$ d.w.) ranged from 980.67 to 1064.37 in leaves, 30.79 to 62.72 in roots, 18.49 to 27.87 in trunks, and 16.40 to 26.32 in shoots. In addition, there were 18.24-fold, 2.21-fold, 1.42-fold, and 1.77-fold increases of B concentrations of leaves, trunks, shoots, and roots of the treated vines compared to control, respectively. However, both own-rooted treated vines and those grafted onto 101-14 Mgt rootstock showed increased concentration of boron in their leaves. Apart from the own-rooted Cabernet Franc and Merlot, no significant differences were observed among the scion varieties. Regarding the other parts of vines, differences between scion cultivars and rootstocks were found significant in terms of root B concentrations, whereas rootstock effects were not significant for shoots and trunks.

### 3.2. Plant Growth Parameters and Toxicity Symptoms

Symptoms of B toxicity manifested on leaf blades of boron-treated vines 25 days after the beginning of the experiment. Smaller leaves, yellowing of the leaf margins and edges, followed by necrosis, were the symptoms of B toxicity. Another specific visible symptom of B toxicity is the downward or upward cupping of young and older leaves, respectively (Figure 1). In our study, the first 101-14 Mgt rootstock expressed toxicity injury symptoms, followed by own-rooted Merlot and Cabernet Franc cultivars. In the following period, the characteristic symptoms appeared in vines grafted onto 101-14 Mgt and 1103P rootstock.

The present experiments demonstrated that B treatments (0.5 mM) significantly affected growth parameters, mainly root and shoot dry weights, whereas trunk weights remained unaffected (Table 3).

After sixty days in toxic B conditions, shoot and root dry weights tended to decrease, although the effect of boron treatment depended on both scion and rootstock cultivars. Analysis of the variance also revealed that the effects of the scion and rootstock cultivar were significant. The Merlot cultivar and vines grafted onto 1103P rootstock showed increased shoot weights. However, when compared to 101-14 Mgt rootstock, control vines

grafted onto 1103P showed increased root weights. No significant rootstock × scion cultivar, rootstock × boron treatment and scion cultivar × boron treatments interactions were found for the shoot and root dry weights.

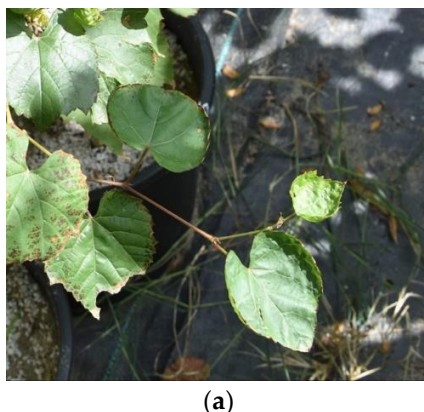 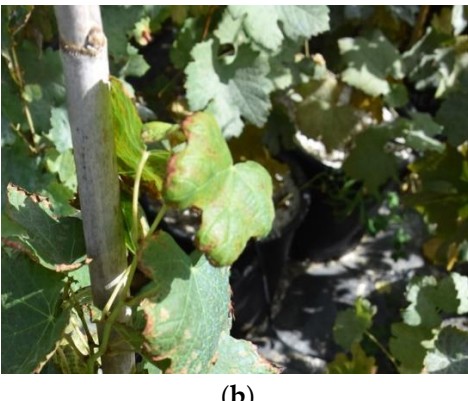

| (a) | (b) |

**Figure 1.** Leaf blade toxicity symptoms twenty-five days after 0.5 mM boron treatment of 101-14 Mgt rootstock (**a**) and own-rooted Merlot (**b**) vines.

**Table 3.** The effect of boron toxicity on root, trunk, and shoot dry weight (g), of own-rooted or grafted onto 1103P and 101-14 Mgt rootstocks of Merlot and Cabernet Franc vine cultivars.

|  |  | Merlot | | | Cabernet Franc | | |
|---|---|---|---|---|---|---|---|
|  |  | Shoot | Trunk | Root | Shoot | Trunk | Root |
| Control | Own roots | 47.88 [b] | 60.51 [a] | 49.73 [a] | 38.91 [d] | 53.11 [c] | 44.98 [b] |
|  | 1103P | 52.15 [a] | 57.76 [b] | 50.93 [a] | 42.45 [c] | 61.01 [a] | 49.30 [a] |
|  | 101-14 Mgt | 42.56 [c] | 58.52 [b] | 41.38 [b] | 38.46 [d] | 61.28 [a] | 39.86 [b] |
| 0.5 mM B | Own roots | 44.24 [bc] | 59.45 [a] | 42.39 [b] | 36.89 [d] | 54.59 [c] | 34.94 [c] |
|  | 1103P | 46.21 [b] | 57.40 [b] | 42.44 [b] | 38.83 [d] | 69.48 [a] | 32.39 [d] |
|  | 101-14 Mgt | 38.46 [d] | 57.62 [b] | 32.39 [d] | 34.08 [d] | 58.39 [bc] | 29.68 [d] |
|  | LSD ($p < 0.05$) | *3.091* | 2.041 | 2.597 | 3.091 | 2.041 | 2.597 |
|  | F < 0.001 | 16.101 | 41.171 | 42.566 | 16.101 | 41.171 | 42.566 |

Different letters in each column represent significant differences at $p < 0.05$.

### 3.3. Tissue Nutrient Concentrations

The concentrations of mineral macronutrients measured in different parts of the vine leaves are shown in Table 4. Analyses of variances among macronutrient (N, P, K, Mg, and Ca) content revealed significant differences due to boron excess. High B concentrations (0.5 mM) significantly decreased N, P, K, Mg, and Ca concentrations.

The concentrations of N, P, K, Mg, and Ca in leaves were significantly decreased by boron toxicity ranging from 15.29% (Ca) to 32 35% (Mg) when compared to control vines. Boron excess also altered the concentrations of mineral nutrients in the roots, decreasing the levels of N, P, K, and Mg ranging from 26.08% (N) to 40.95% (K), but without having a significant impact on Ca. Regarding the other parts, boron-treated vines showed significantly lower macronutrient concentrations except for shoot P and trunk Ca. The leaves of the vines had the highest concentrations of macronutrients, whereas the trunks exhibited the lowest levels of N, P, K, and Mg. Own-rooted vines recorded higher K concentrations in leaves compared to grafted vines, according to the research. In addition, compared to 1103P, the vines grown onto 101-14 Mgt rootstock maintained higher K levels in their leaves. No significant rootstock × boron, Scion variety × rootstock, and Scion variety × boron interactions were found for the K concentrations in leaves. A similar effect was found for leaf Ca and Mg concentrations, with own-rooted vines or grafted onto 1103P rootstock showing increased concentrations in leaves compared to those grafted onto 101-14 Mgt.

Regarding the scion and rootstock varieties, no significant differences were found in root macronutrient concentrations except for K, whereas the own-rooted vines showed higher values compared to the grafted vines. Compared to grafted vines, own-rooted vines showed higher K values in shoots. In addition, there were no significant differences concerning the trunk macronutrients between the scion and rootstock varieties.

**Table 4.** The effect of boron toxicity on N, P, K, Mg and Ca (% d.w.) concentration in different parts of Merlot and Cabernet Franc vines on own roots or grafted to 1103P and 101-14 Mgt rootstocks.

| | | | | Leaves | | | | | | | |
|---|---|---|---|---|---|---|---|---|---|---|---|
| | | | Merlot | | | | | Cabernet Franc | | | |
| Treatments | | N | P | K | Mg | Ca | N | P | K | Mg | Ca |
| Control | Own roots | 2.16 [a] | 0.30 [b] | 2.35 [a] | 0.61 [a] | 2.88 [a] | 2.23 [a] | 0.33 [a] | 2.24 [a] | 0.62 [a] | 2.96 [a] |
| | 1103P | 2.44 [a] | 0.32 [a] | 1.33 [c] | 0.72 [a] | 2.67 [a] | 1.95 [b] | 0.32 [a] | 1.65 [c] | 0.63 [a] | 2.88 [a] |
| | 101-14 Mgt | 1.77 [b] | 0.27 [b] | 1.56 [c] | 0.56 [b] | 2.63 [ab] | 1.94 [b] | 0.29 [b] | 1.96 [b] | 0.55 [b] | 2.70 [a] |
| 0.5 mM B | Own roots | 1.83 [ab] | 0.23 [c] | 0.97 [d] | 0.42 [c] | 2.45 [bc] | 1.89 [b] | 0.28 [b] | 1.42 [c] | 0.44b [c] | 2.63 [ab] |
| | 1103P | 1.81 [ab] | 0.20 [c] | 1.26 [c] | 0.49 [b] | 2.29 [c] | 1.77 [b] | 0.27 [b] | 1.47 [c] | 0.45 [bc] | 2.46 [b] |
| | 101-14 Mgt | 1.68 [b] | 0.25b [c] | 1.75 [bc] | 0.31 [d] | 1.96 [c] | 1.74 [b] | 0.22 [c] | 1.71 [bc] | 0.39 [c] | 2.37 [b] |
| | LSD (*p* < 0.05) | 0.177 | 0.024 | 0.249 | 0.122 | 0.158 | 0.177 | 0.024 | 0.249 | 0.122 | 0.158 |
| | F < 0.001 | 4.845 | 7.398 | 10.73 | 10.227 | 8.558 | 4.845 | 7.398 | 10.73 | 10.227 | 8.558 |
| | | | | Roots | | | | | | | |
| | | | Merlot | | | | | Cabernet Franc | | | |
| Treatments | | N | P | K | Mg | Ca | N | P | K | Mg | Ca |
| Control | Own roots | 1.14 [a] | 0.29 [a] | 0.58 [a] | 0.28 [a] | 1.18 | 1.16 [a] | 0.27 [a] | 0.65 [a] | 0.32 [a] | 1.16 |
| | 1103P | 1.12 [a] | 0.30 [a] | 0.50 [a] | 0.31 [a] | 1.29 | 1.17 [a] | 0.29 [a] | 0.48 [a] | 0.31 [a] | 1.11 |
| | 101-14 Mgt | 1.20 [a] | 0.29 [a] | 0.52 [a] | 0.32 [a] | 1.03 | 1.12 [a] | 0.28 [a] | 0.42 [b] | 0.38 [a] | 1.05 |
| 0.5 mM B | Own roots | 0.79 [b] | 0.18 [b] | 0.30 [b] | 0.21 [b] | 1.24 | 0.90 [a] | 0.20 [b] | 0.39 [b] | 0.21 [b] | 0.94 |
| | 1103P | 0.88 [ab] | 0.20 [b] | 0.31 [b] | 0.20 [b] | 0.95 | 0.83 [b] | 0.19 [b] | 0.36 [b] | 0.23 [b] | 0.98 |
| | 101-14 Mgt | 0.83 [b] | 0.22 [ab] | 0.26 [b] | 0.23 [b] | 0.97 | 0.88 [ab] | 0.23 [ab] | 0.24 [b] | 0.19 [b] | 0.90 |
| | LSD (*p* < 0.05) | 0.321 | 0.094 | 0.069 | 0.106 | ns | 0.321 | 0.094 | 0.069 | 0.106 | ns |
| | F< 0.001 | 9.950 | 22.627 | 8.755 | 8.825 | | 9.950 | 22.627 | 8.755 | 8.825 | |
| | | | | Shoots | | | | | | | |
| | | | Merlot | | | | | Cabernet Franc | | | |
| Treatments | | N | P | K | Mg | Ca | N | P | K | Mg | Ca |
| Control | Own roots | 1.06 [a] | 0.23 | 0.66 [a] | 0.22 [b] | 0.76 [a] | 0.95 [b] | 0.19 | 0.61 [b] | 0.18 [c] | 0.67 [ab] |
| | 1103P | 1.07 [a] | 0.18 | 0.59 [b] | 0.19 [c] | 0.77 [a] | 1.05 [a] | 0.20 | 0.67 [a] | 0.24 [a] | 0.70 [a] |
| | 101-14 Mgt | 1.09 [a] | 0.18 | 0.60 [b] | 0.25 [a] | 0.70 [a] | 0.98 [b] | 0.21 | 0.52 [c] | 0.21 [b] | 0.76 [a] |
| 0.5 mM B | Own roots | 0.96 [b] | 0.19 | 0.34 [d] | 0.14 [c] | 0.65 [b] | 0.86 [c] | 0.16 | 0.40 [c] | 0.15 [c] | 0.61 [b] |
| | 1103P | 0.89 [c] | 0.15 | 0.45 [c] | 0.16 [c] | 0.64 [b] | 0.79 [c] | 0.20 | 0.29 [d] | 0.16 [c] | 0.59 [b] |
| | 101-14 Mgt | 0.79 [c] | 0.16 | 0.36 [d] | 0.15 [c] | 0.60 [b] | 0.75 [c] | 0.17 | 0.28 [d] | 0.14 [c] | 0.66 [b] |
| | LSD (*p* < 0.05) | 0.071 | ns | 0.058 | 0.029 | 0.10 | 0.071 | ns | 0.058 | 0.029 | 0.10 |
| | F < 0.001 | 5.752 | | 23.831 | 16.85 | 7.360 | 5.752 | | 23.831 | 16.85 | 7.360 |
| | | | | Trunks | | | | | | | |
| | | | Merlot | | | | | Cabernet Franc | | | |
| Treatments | | N | P | K | Mg | Ca | N | P | K | Mg | Ca |
| Control | Own roots | 0.78 [a] | 0.25 [a] | 0.25 [a] | 0.15 [b] | 0.86 | 0.76 [a] | 0.24 [a] | 0.27 [a] | 0.16 [b] | 0.80 |
| | 1103P | 0.81 [a] | 0.22 [a] | 0.28 [a] | 0.16 [b] | 1.52 | 0.80 [a] | 0.14 [ab] | 0.25 [a] | 0.14 [b] | 0.82 |
| | 101-14 Mgt | 0.55 [b] | 0.16 [ab] | 0.27 [a] | 0.18 [a] | 0.70 | 0.74 [a] | 0.25 [a] | 0.24 [a] | 0.16 [b] | 0.85 |
| 0.5 mM B | Own roots | 0.62 [ab] | 0.08 [b] | 0.17 [b] | 0.12 [c] | 0.62 | 0.58 [b] | 0.11 [b] | 0.18 [b] | 0.13 [c] | 0.64 |
| | 1103P | 0.57 [b] | 0.10 [b] | 0.18 [b] | 0.13 [c] | 0.70 | 0.54 [b] | 0.09 [b] | 0.17 [b] | 0.11 [c] | 0.72 |
| | 101-14 Mgt | 0.60 [b] | 0.09 [b] | 0.16 [b] | 0.12 [c] | 0.61 | 0.64 [ab] | 0.11 [b] | 0.16 [b] | 0.12 [c] | 0.58 |
| | LSD (*p* < 0.05) | 0.197 | 0.088 | 0.091 | 0.013 | ns | 0.197 | 0.088 | 0.091 | 0.013 | ns |
| | F < 0.001 | 8.053 | 2.460 | 8.061 | 17.916 | | 8.053 | 2.460 | 8.061 | 17.916 | |

Ns: not significant. Different letters in each column represent significant differences at *p* < 0.05.

In the case of micronutrients (Table 5), boron treatment significantly affected all nutrient concentrations studied in our experiment. Iron was the most concentrated micronutrient present in the leaf and root tissues, reaching 107.50 mg kg$^{-1}$ and 208.46 mg kg$^{-1}$, respectively. Overall, all the measured micronutrient elements accumulated in various tissues decreased after boron application ranging from 17.92% to 46.28% for Zinc in leaves and roots, respectively. According to the results obtained, excess B decreased iron concentration from 19.53% to 38.93% in leaves and roots, respectively. A similar decrease was observed for manganese (31.64% roots) and copper (40.20% roots). No significant differences were found between the rootstocks for leaf iron concentrations. On the contrary, leaves of the Cabernet Franc cultivar showed increased leaf iron concentrations compared to merlot leaves. Furthermore, the Mn, Zn, and Cu concentrations in the leaf were significantly affected by the rootstock cultivar. As presented in Table 5, root, shoot and trunk Mn concentrations were markedly affected by excess B, decreasing by 31.84%, 23.48%, and 23.43%, respectively. Likewise, corresponding decreases of 46.28%, 43.97%, and 31.51% for Zn and 40.20%, 26.72% and 29.58 for Cu were found.

**Table 5.** The effect of boron toxicity on Mn, Zn, Fe and Cu (mg kg$^{-1}$) concentration in different parts of Merlot and Cabernet Franc vines on their own roots or grafted onto 1103P and 101-14 Mgt rootstocks.

| | | Leaves | | | | | | | |
|---|---|---|---|---|---|---|---|---|---|
| | | Merlot | | | | Cabernet Franc | | | |
| Treatments | | Mn | Zn | Fe | Cu | Mn | Zn | Fe | Cu |
| Control | Own roots | 97.70 [a] | 16.25 [b] | 114.34 [a] | 10.60 [a] | 68.02 [b] | 14.06 [b] | 112.26 [b] | 9.15 [a] |
| | 1103P | 96.03 [a] | 19.03 [a] | 106.21 [b] | 8.98 [a] | 54.19 [b] | 15.05 [b] | 135.68 [a] | 7.61 [a] |
| | 101-14 Mgt | 55.03 [b] | 14.31 [b] | 125.10 [a] | 7.51 [ab] | 50.09 [b] | 15.43 [b] | 121.26 [a] | 7.01 [ab] |
| 0.5 mM B | Own roots | 59.54 [b] | 13.51 [b] | 95.94 [b] | 6.78 [b] | 57.01 [b] | 10.92 [bc] | 99.14 [b] | 5.67 [b] |
| | 1103P | 56.85 [b] | 15.29 [b] | 95.66 [b] | 5.98 [b] | 47.59 [bc] | 14.32 [b] | 96.34 [b] | 5.45 [b] |
| | 101-14 Mgt | 41.94 [bc] | 12.80 [b] | 99.18 [b] | 5.02 [c] | 37.13 [c] | 10.39 [c] | 89.01 [b] | 5.87 [b] |
| | LSD (*p* < 0.05) | 17.619 | 1.988 | 22.03 | 2.11 | 17.619 | 1.988 | 22.03 | 2.11 |
| | F < 0.001 | 30.316 | 14.816 | 10.907 | 10.712 | 30.316 | 14.816 | 10.907 | 10.712 |
| | | **Roots** | | | | | | | |
| | | Merlot | | | | Cabernet Franc | | | |
| Treatments | | Mn | Zn | Fe | Cu | Mn | Zn | Fe | Cu |
| Control | Own roots | 32.68 [a] | 28.77 [b] | 243.28 [a] | 28.13 [a] | 35.14 [a] | 26.12 [b] | 238.91 [a] | 30.60 [ab] |
| | 1103P | 31.90 [a] | 37.76 [a] | 276.31 [a] | 32.68 [a] | 34.18 [a] | 25.10 [b] | 251.01 [a] | 33.45 [abc] |
| | 101-14 Mgt | 34.96 [a] | 35.83 [a] | 302.65 [a] | 30.71 [a] | 31.61 [a] | 32.31 [b] | 246.51 [a] | 35.01 [a] |
| 0.5 mM B | Own roots | 21.43 [b] | 15.19 [c] | 159.88 [b] | 19.50 [ab] | 24.30 [a] | 14.94 [c] | 160.74 [b] | 19.20 [c] |
| | 1103P | 22.95 [b] | 18.96 [c] | 138.36 [b] | 18.04 [b] | 23.24 [b] | 17.93 [c] | 176.11 [b] | 18.58 [c] |
| | 101-14 Mgt | 19.98 [b] | 16.12 [c] | 146.35 [b] | 20.81 [ab] | 24.73 [b] | 16.75 [c] | 170.57 [b] | 17.82 [bc] |
| | LSD (*p* < 0.05) | 10.63 | 3.93 | 100.56 | 12.97 | 10.63 | 3.93 | 100.56 | 12.97 |
| | F < 0.001 | 17.38 | 16.945 | 7.313 | 9.942 | 17.38 | 16.945 | 7.313 | 9.942 |
| | | **Shoots** | | | | | | | |
| | | Merlot | | | | Cabernet Franc | | | |
| Treatments | | Mn | Zn | Fe | Cu | Mn | Zn | Fe | Cu |
| Control | Own roots | 39.57 [a] | 26.80 [a] | 24.87 [a] | 11.51 [a] | 29.01 [b] | 23.09 [a] | 25.16 [a] | 10.32 [a] |
| | 1103P | 38.92 [a] | 20.52 [a] | 25.15 [a] | 10.55 [a] | 37.71 [b] | 30.14 [a] | 23.42 [a] | 11.35 [a] |
| | 101-14 Mgt | 40.20 [a] | 27.71 [a] | 23.89 [a] | 11.57 [a] | 35.13 [ab] | 19.12 [b] | 23.81 [a] | 10.74 [a] |

**Table 5.** *Cont.*

| | | Leaves | | | | | | | |
|---|---|---|---|---|---|---|---|---|---|
| | | Merlot | | | | Cabernet Franc | | | |
| **Treatments** | | **Mn** | **Zn** | **Fe** | **Cu** | **Mn** | **Zn** | **Fe** | **Cu** |
| | Own roots | 24.29 [b] | 14.15 [b] | 16.50 [b] | 8.28 [b] | 26.24 [b] | 13.62 [b] | 21.31 [a] | 8.48 [b] |
| 0.5 mM B | 1103P | 31.62 [ab] | 13.16 [b] | 16.28 [b] | 8.10 [b] | 27.81 [b] | 15.57 [b] | 15.24 [b] | 8.84 [b] |
| | 101-14 Mgt | 37.50 [a] | 15.60 [b] | 15.24 [b] | 7.31 [b] | 21.24 [b] | 11.66 [b] | 16.87 [b] | 7.36 [b] |
| | LSD (*p* < 0.05) | 4.239 | 10.68 | 7.62 | 3.26 | 4.239 | 10.68 | 7.62 | 3.26 |
| | F < 0.001 | 16.976 | 31.49 | 7.851 | 12.921 | 16.976 | 31.49 | 7.851 | 12.921 |
| | | Trunks | | | | | | | |
| | | Merlot | | | | Cabernet Franc | | | |
| **Treatments** | | **Mn** | **Zn** | **Fe** | **Cu** | **Mn** | **Zn** | **Fe** | **Cu** |
| | Own roots | 26.21 [a] | 17.08 [a] | 85.11 [b] | 8.83 [ab] | 20.49 [b] | 19.10 [a] | 81.23 [b] | 10.15 [a] |
| Control | 1103P | 26.02 [a] | 15.97 [a] | 106.88 [a] | 10.28 [a] | 24.88 [a] | 17.55 [a] | 94.13 [ab] | 10.12 [a] |
| | 101-14 Mgt | 24.16 [a] | 17.71 [a] | 97.31 [a] | 9.41 [a] | 24.20 [a] | 18.08 [ab] | 95.51 [a] | 9.83 [a] |
| | Own roots | 17.97 [b] | 11.01 [b] | 60.58 [cd] | 6.25 [b] | 17.97 [b] | 10.76 [b] | 49.93 [d] | 8.90 [ab] |
| 0.5 mM B | 1103P | 18.57 [b] | 10.85 [b] | 83.40 [b] | 6.23 [b] | 18.56 [b] | 11.51 [b] | 60.63 [c] | 6.98 [b] |
| | 101-14 Mgt | 19.87 [b] | 10.07 [b] | 71.10 [c] | 5.85 [b] | 19.08 [b] | 13.51 [ab] | 70.03 [c] | 7.10 [b] |
| | LSD (*p* < 0.05) | 1.69 | 4.683 | 10.046 | 3.09 | 1.69 | 1.69 | 10.046 | 3.09 |
| | F < 0.001 | 17.694 | 1.242 | 18.406 | 11.204 | 17.694 | 17.694 | 18.406 | 11.204 |

Different letters in each column represent significant differences at *p* < 0.05.

### 3.4. Chlorophyll Content

Chlorophyll was extracted from leaves and measured to investigate the changes on the 1st, 30th and 60th day after exposure to boron excess. At the same time, the chlorophyll index CCM-200 was also recorded.

Figure 2 demonstrates the effect of boron toxicity on total chlorophyll content and CCM-200 index over the experimentation period. The total leaf chlorophyll concentration significantly declined from 35.46% to 44.45%, thirty and sixty days, respectively, after the beginning of treatments, due to high B levels. The general trend for the treatment during successive measurement periods (after 30 and 60 days of treatment) shows a gradual reduction in the content. In addition, neither the rootstock nor the scion cultivar has a significant effect on the total chlorophyll. On the 60th day of the experimentation, both grafted and own-rooted vines showed the lowest chlorophyll concentrations. The relative chlorophyll content (CCM-200) shows a similar trend. The minimum value of chlorophyll index (CCM-200) (16.16) was found in treated, own-rooted Cabernet Franc vines after sixty days of experimentation.

### 3.5. Total Phenolics

The total phenolic contents in the examined extracts ranged from 28.98 to 57.68 mg g$^{-1}$ f.w. (CE) (Figure 3). The application of boron excess significantly increased phenolic concentrations from 71.03% in Merlot vines to 73.39% in Cabernet Franc. Analysis of variance revealed no significant effect of rootstock and scion cultivar variety.

### 3.6. Water Status and Photosynthetic Activity

As shown in Figure 4, stem water potential (SWP), measured at mid-day, showed a gradual decrease over the growing season. A strong decrease in SWP was observed from the initial stage to sixty days of experimentation. All the SWP values in treated vines were comparable to control values on day one. Significant differences were observed among the three measurement stages (LSD *p* < 0.05: 0.366). Since no significant interactions between the different stages and boron treatment were observed, it was possible to evaluate the progression of SWP separately as functions of two factors of variability. After 60 days of

experimentation, SWP decreased from −0.51 MPa in control vines to −1.36 in boron-treated ones. An excessive supply of boron significantly decreased SWP (LSD $p < 0.05$: 0.076, F: 58.612). Vines exposed to 0.5 mM boron for 60 days showed lower SWP compared to the control. Both the rootstock cultivar and the boron treatment were found to have significant impacts, according to the analysis of variance. No significant rootstock cultivar × boron treatment interactions were found. Additionally, when compared to 1103P, 101-14 Mgt rootstock consistently exhibited lower stem water potential values (LSD $p < 0.05$: 0.052, F: 58.612).

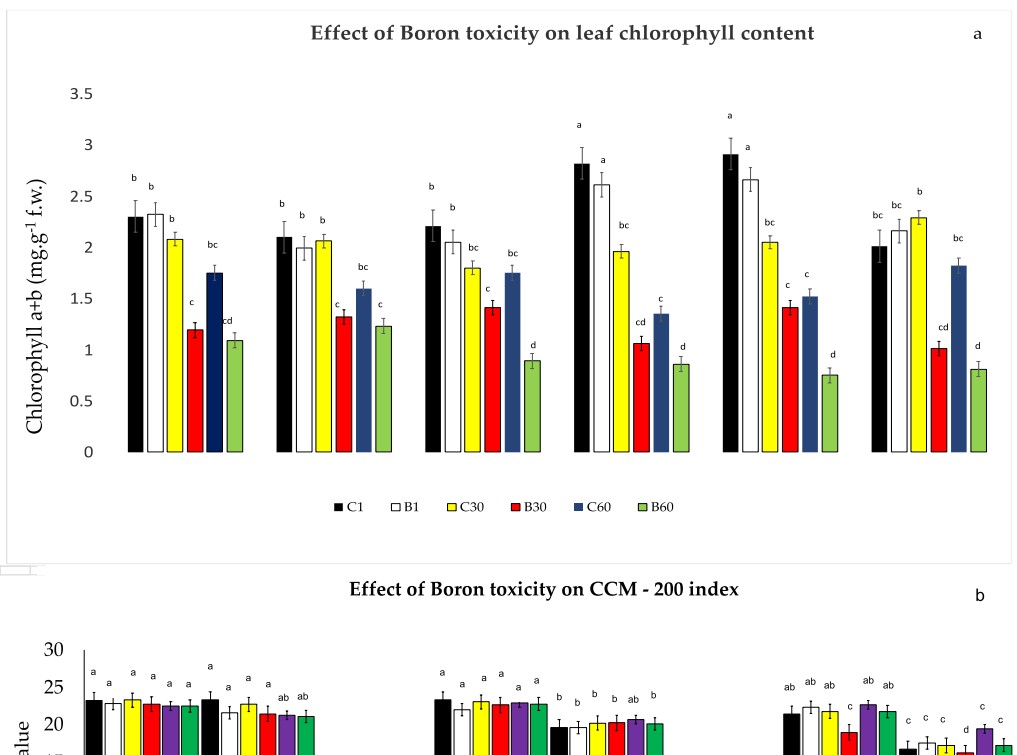

**Figure 2.** Leaf chlorophyll content (**a**) and CCM-200 Chlorophyll index (**b**) at the 1st, 30th and 60th day from the beginning of boron treatment of own-rooted (OR) or grafted onto 1103P and 101-14 Mgt, Merlot (M) and Cabernet Franc (CF) vine varieties. C represents the control treatment; B represents the boron treatment. Different letters represent significant differences at $p < 0.05$.

The photosynthetic activity of vines exposed to excess boron showed significant changes in terms of net $CO_2$ assimilation rate (A), and PSII chlorophyll fluorescence (ChF).

Figure 5 presents the variation of A and gs with excess boron in vines that were either own-rooted or grafted onto two different rootstocks after thirty and sixty days of treatment. Throughout the experimental period, measurements of A and gs were taken in different plots. Significant differences in gs and A were found among the two stages (LSD $p < 0.05$: 0.041, F: 2.436 and 5.52, 14.673, respectively). Our results showed that the net $CO_2$ assimilation rate decreased, ranging from 5.42 to 10.50 μmol $CO_2$ m$^{-2}$ s$^{-1}$ on day thirty and from 2.76 to 6.90 on the final day of the treatment cycle.

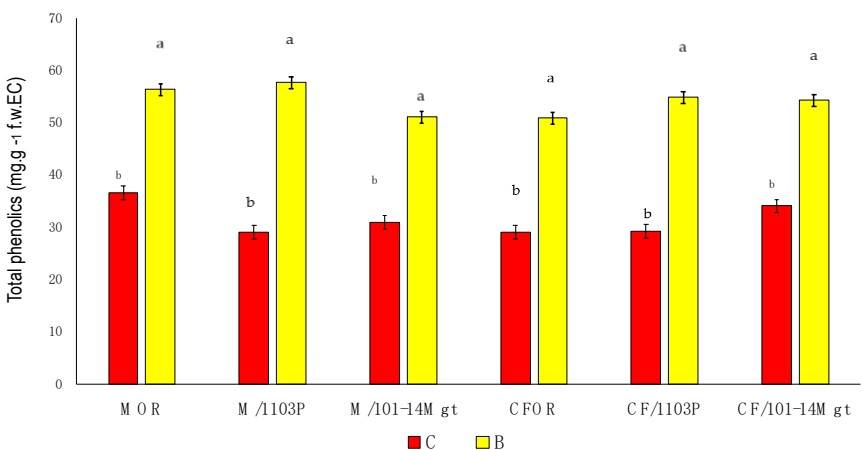

**Figure 3.** Leaf total phenolic content in Merlot (M) and Cabernet Franc (CF) vine varieties, either own-rooted (OR) or grafted onto 1103P and 101-14 Mgt rootstocks after sixty days of boron treatment. C represents the control treatment; B represents the borontreatment. Different letters represent significant differences at *p* < 0.05.

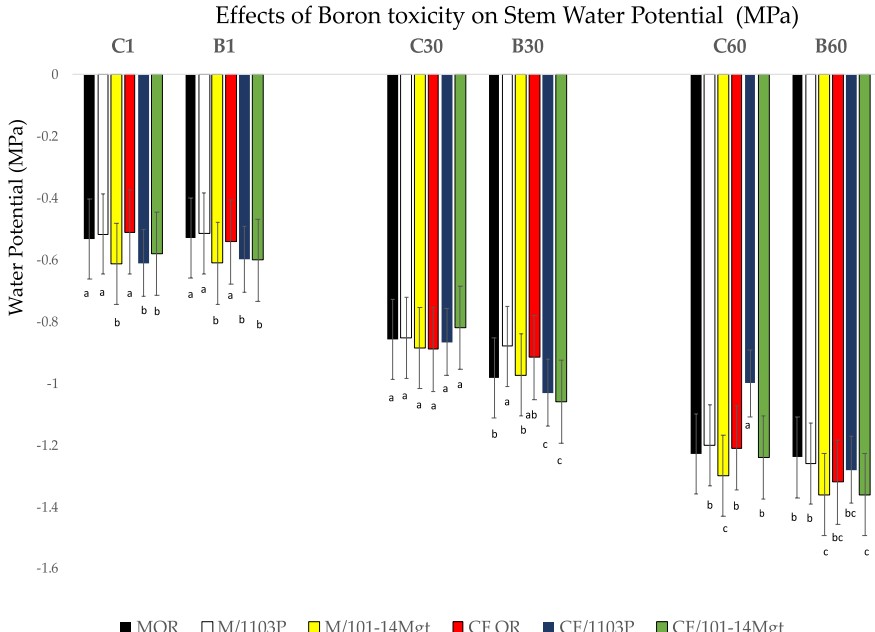

**Figure 4.** Boron toxicity effects on midday stem water potential of Merlot (M) and Cabernet Franc (CF) grapevine cultivars on their own roots (OR) and on 1103P and 101-14 Mgt rootstocks at three stages. C represents the control treatment; B represents the boron treatment. 1, 30, 60 (at day one, thirty and sixty, respectively). Different letters represent significant differences at *p* < 0.05.

In addition, plots treated with 0.5 mM boron showed decreased photosynthetic activity. Significantly reduced values of A were observed in boron-treated plots (LSD *p* < 0.05: 1.45 F: 14.673), whereas stomatal conductance was not affected. However, the experimental duration significantly affected stomatal conductance, which decreased by up to 30% between day thirty and day sixty. At the same time, the net $CO_2$ assimilation rate decreased by 19.64 and 48.54% in control and treated vines, respectively. Furthermore, there were notable variations in the net $CO_2$ assimilation rate between the scion cultivars and the rootstocks used in the experiment. Generally, the Merlot cultivar and 1103P rootstock showed an increased net $CO_2$ assimilation rate. When grafted onto 101-14 Mgt rootstock sixty days following the beginning of the boron treatment, the Cabernet Franc cultivar

attained the lowest values of net $CO_2$ assimilation rate and gs (2.76 μmol $CO_2$ m$^{-2}$ s$^{-1}$ and 0.08 mol m$^{-2}$ s$^{-1}$ respectively).

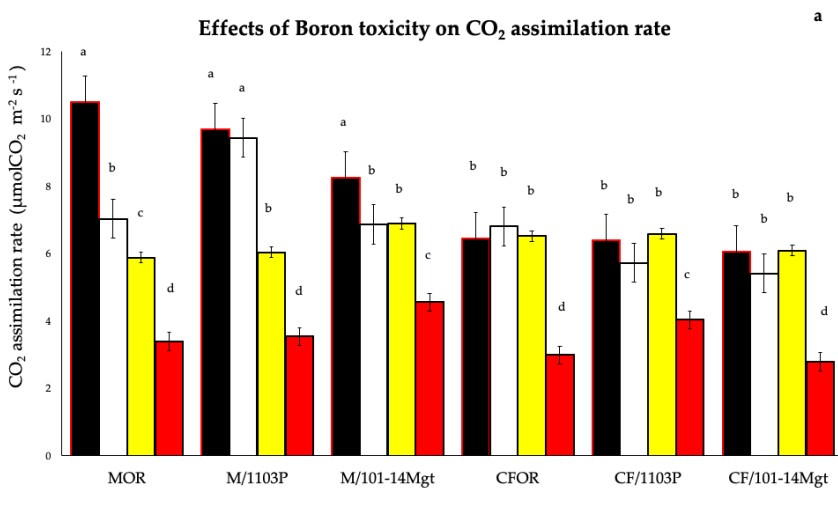

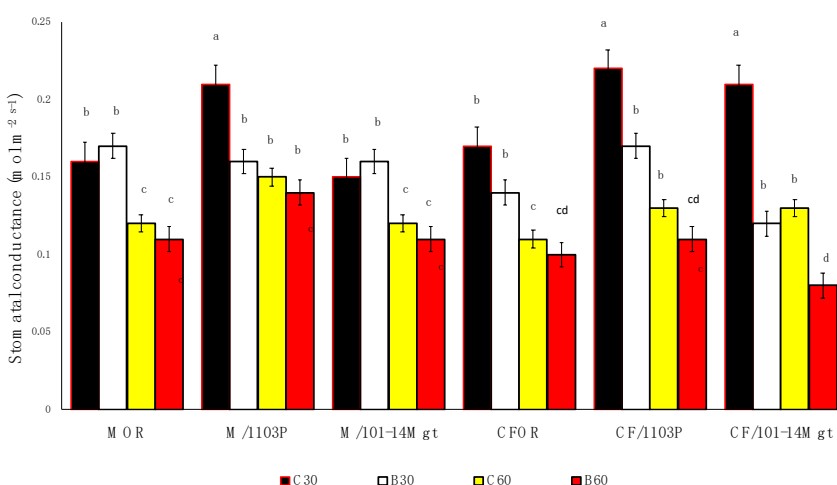

**Figure 5.** Net $CO_2$ assimilation rate (**a**) and stomatal conductance (**b**) of Merlot (M) and Cabernet Franc (CF) grapevine cultivars on their own roots (OR) and on 1103P and 101-14 Mgt rootstocks, thirty and sixty days after the start of boron treatments. C represents the control treatment; B represents the boron treatment. 30, 60 (at day thirty and sixty, respectively). Different letters represent significant differences at $p < 0.05$.

Chlorophyll fluorescence parameters were measured in dark-adapted leaves, and $F_v/F_m$ values related to the maximum quantum yield of PS II were calculated. Between the two stages of measurement and between the treatment with or without boron, there were significant differences in the $F_v/F_m$ ratio. Moreover, cultivar and rootstock effects, as well as rootstock × boron interactions were significant. $F_v/F_m$ gradually decreased over the experimental period, reaching the lowest values sixty days following the beginning of treatment with excess boron, showing the lowest values in the own-rooted Cabernet Franc cultivar.

## 4. Discussion

Water relations, growth and development, leaf gas exchange, nutrient uptake, and transport are just a few of the developmental or biochemical processes that may be affected

by boron toxicity in plants. B toxicity in grapevines is usually evident by characteristic symptoms on above-ground parts, including decreased shoot growth, smaller leaves, and also marginal chlorosis and necrosis.

### 4.1. Vine Growth and Nutrient Concentrations in Plant Tissues

For both ungrafted and grafted vines, the boron concentrations in each plant organ increased with excess boron treatment (Table 2). Among the different vine organs, leaves showed the highest boron concentration, followed by roots and other parts. It has been reported that in most plant species, boron is absorbed by roots, loaded into the xylem, and translocated to shoots via the transpiration stream and accumulated in leaves without undergoing redistribution [29]. However, in relatively lower concentrations, boron can also be found accumulated in roots. Dannel et al., 1998 [30] reported that after long-term experiments with excess B, the concentration in sunflower leaves was about 10-fold higher than in the roots, indicating that roots do not retain B to prevent the above-ground from accumulating excess B. These tissues typically contain between 40 and 100 mg B kg$^{-1}$ d.w. in species that accumulate B in their leaves. However, when B levels in the soil approach toxic levels, the leaves may contain 250 mg kg$^{-1}$ d.w. B. Under extremely toxic B conditions, leaf B concentrations may exceed 700 to 1000 mg kg$^{-1}$ d.w. [31]. In grapevines, similar symptoms and decreased shoot vigor have been reported in field conditions when leaf B concentrations were greater than 140 mg kg$^{-1}$ d.w. [22].

Boron is absorbed by plants primarily passively as boric acid but also in small amounts actively as borate ions [3]. A system of active transportation using B transporters has been suggested in the scenario of restricted B availability. In Arabidopsis, BOR1 was reported to be the initially identified transporter involved in xylem loading [32]. In B-limiting conditions, a mechanism involving various protein channels and transporters is established to supply this component to aerial tissues [33]. Transcriptional analysis of a boron transporter gene (VvBOR1) in the grapevine revealed it is preferentially expressed in flowers at anthesis, and there is a direct correlation between the expression pattern and the amount of boron in grapes [34,35].

The results of the present research are in accordance with those presented in citrus [36] and mandarin [37], where B accumulated primarily in the leaves as opposed to other plant tissues. The increased boron concentrations in leaves as a result of an excess of B application is a common and expected response. However, in our experiment, significant differences were found between the different rootstocks. Both the own-rooted treated vines and vines grafted onto 101-14 Mgt rootstock concentrated increased boron in leaves, resulting in the earlier appearance of leaf toxicity symptoms in vines grafted onto 101-14 Mgt rootstock, in comparison with 1103P.

Macronutrient levels in leaves were adequate for vine growth since N, P, K, Mg and Ca concentrations in leaves during this experiment ranged between 1.68 and 2.44%, 0.20 and 0.32%, 0.97 and 3.35%, 0.42 and 0.72%, 1.96 and 2.88% in d.w., respectively. However, excess boron significantly decreased macronutrients in various parts of the vine in most cases. These results are consistent with those of other researchers who reported decreased leaf nutrients because of a high B supply [22,38]. Moreover, it has been reported that P concentration was decreased by high B in grapevines [22] as well as in kiwi fruit [39] and tomato [40].

Ca and Mg are crucial for the development and growth of plants. Ca crosslinks the carboxyl groups of the pectic polymers to enhance the cell wall, whereas Mg is a crucial component of proteins and chlorophyll [41]. The distribution of Mg was generally more concentrated in leaves, indicating its high mobility and transport efficiency in plants.

According to our results, all measured trace elements accumulated in various tissues decreased after boron application. Fe content in vine tissues was significantly higher compared to the other elements, which may be attributed to the plant's absorption mechanism and functions. Iron participates in redox reactions, electron transport, and maintaining the chloroplast's structural integrity in cells to ensure that plants can absorb and distribute

sufficient Fe to leaves in various environments [42]. Furthermore, we noticed a significant decline in the uptake of Fe in grapevine plants exposed to excess boron. The results clearly demonstrate that B treatment reduces Cu, Mn, and Zn concentrations in various vine parts. Although Cu and Zn are contaminants at certain concentrations, they are also essential micronutrients for plants. For instance, Zn plays a role as an activator of enzymes, and Cu plays an important role in regulating protein composition, photosynthesis, mitochondrial respiration, and cell wall metabolism [43]. In the current study, Cu and Zn were equally distributed throughout all tissues, while Cu was more concentrated in roots. The decrease in C uptake by roots could be due to the toxic effect of B on root cells, leading to an impaired absorption process. The low uptake of Cu and Zn by plants was attributed to excess boron, according to Singh et al., 1990 [44]. However, previous studies did not show consistent effects of excess boron on nutrient elements of different plant species [45–47]. These inconsistencies between various studies could be caused by differences in the mobility of nutrients in different species or variations in the demands of these crops for nutrients throughout the growing period [45].

In our experiment, the first visible symptoms of B toxicity developed on leaf blades of B-treated vines 25 days after the beginning of the experimentation (Figure 1).

It has been reported that after being absorbed, boron is transported to the shoot via transpiration flux, and tends to accumulate radially at the leaf edges where leaf veins end. These tissues indeed show the most severe symptoms of B toxicity [48]. In grapevines, which have leaf reticulate venation, B toxicity is observed around the leaf margins, whereas in grasses, such as wheat and barley, with parallel-veined leaves, the toxic effect develops black patches in leaf tips where the veins terminate [49]. The disorder was also characterized by the downward or upward cupping of leaves. Occasionally, leaf cupping has been reported as a symptom of B toxicity [50,51]. According to Loomis and Durst 1992 [52], the disorder may be induced by the inhibition of cell growth, possibly due to an abnormal high level of cross-links in the cell wall. B toxicity symptoms do not appear in roots, which suggests that B distribution is related to the transpiration stream. B commonly accumulated in the leaves, but it remained in the root system when B was in relatively lower concentrations. Although there were no visible symptoms in roots under excess B, the overall root volume significantly decreased (Table 3). A phenotypic effect of B toxicity was related to the inhibition of root growth, which was accompanied by a reduction in plant dry weight and an increase in B levels in root tissues. The reduction of root growth has been previously observed in grapevines by Gunes et al., 2006 [21], and also by others in different plant species [53,54]. Additionally, Aquea et al., 2012 [55] revealed the molecular basis of root growth inhibition caused by B toxicity in *Arabidopsis*. They revealed that B toxicity induced the expression of genes associated with cell wall modifications, abscisic acid response, and abscisic acid signaling. In this study, the decreased leaf chlorophyll content because of increased boron levels led to decreased dry weights of shoots and roots. Regarding the rootstock cultivar effect, it was found that vines grafted onto 1103P showed increased root and shoot weights compared to those grafted onto 101-14 Mgt rootstock.

### 4.2. Photosynthetic Activity, Water Status, and Chlorophyll Pigments

Excess boron treatments gradually decreased photosynthesis, simultaneously with a decrease in leaf chlorophyll. The damage to the photosynthetic apparatus constitutes the most significant consequence of excess boron. In our experiment, leaf chlorophyll (Chl) content and CCM-200 index dropped thirty days after the beginning of boron treatment (Figure 2). It was reported that plants under boron stress conditions presented a decreased amount of Chl and, therefore, decreases in photosynthetic rate occurred [56,57]. The strong decline of Chl induced by B toxicity in grapevines, as reported in other species [56,58], can indicate a decrease in the synthesis of these molecules, as well as an enhancement of their oxidative processes. It is also well-known that during the late stages of the experiment, plants regulate the construction and destruction of a specific subset of light-harvesting complexes through the formation and degradation of light-reduced total leaf Chl concentrations.

According to Kiani-Pouya and Rasouli 2014 [59], the relative Chl measurement (CCM-200 index) could be used for a cost-effective and rapid chlorophyll assessment because of a close correlation between the CCM-200 index and leaf Chl content. The vines exposed to excess boron during a period of thirty or sixty days showed a decreased photosynthetic activity. At the beginning of the experimental cycle, there were no statistically significant differences for the control plants. To better comprehend the impact of excess B on PSII machinery, we measured the chlorophyll fluorescence parameter $F_v/F_m$ (Table 6). The most frequently used fluorescence parameter is $F_v/F_m$ (maximum quantum yield of PSII). In our study, the maximum quantum yield of PSII was affected by B (Table 6), indicating serious damage to PSII machinery. Similar findings were obtained by Papadakis et al., 2004 [37], who reported a significant decrease in $F_v/F_m$ in leaves of Navelina orange plants grown with excess B concentrations. Our results indicated that both control and stressed vines had relatively high $F_v/F_m$ ratios after 30 days of boron treatment (0.729–0.828). Strong and significant differences between stressed and control vines were observed during the following period, especially at the end of the experimental cycle (60 d). At this stage, Cabernet Franc grafted onto 101-14 Mgt rootstock recorded the lowest value of $F_v/F_m$ ratio (0.489).

**Table 6.** Effects of boron toxicity on the maximum quantum yield of photosystem II ($F_v/F_m$) in own-rooted Merlot and Cabernet Franc grapevine cultivar or grafted onto 1103P and 101-14 Mgt rootstocks.

| | | Merlot | | Cabernet Franc | |
|---|---|---|---|---|---|
| | | **Day Thirty** | **Day Sixty** | **Day Thirty** | **Day Sixty** |
| Control | Own roots | 0.758 [b] | 0.725 [b] | 0.780 [ab] | 0.661 [c] |
| | 1103P | 0.815 [ab] | 0.722 [b] | 0.774 [ab] | 0.705 [bc] |
| | 101-14 Mgt | 0.828 [a] | 0.788 [ab] | 0.818 [a] | 0.727 [b] |
| 0.5 mMB | Own roots | 0.768 [b] | 0.505 [d] | 0.783 [ab] | 0.489 [d] |
| | 1103P | 0.796 [ab] | 0.729 [b] | 0.788 [ab] | 0.593 [c] |
| | 101-14 Mgt | 0.790 [ab] | 0.644 [c] | 0.729 [b] | 0.543 [c] |
| | | LSD ($p < 0.05$): 0.044, F < 0.001: 19.057 | | | |

Different letters in each column represent significant differences at $p < 0.05$.

The significant decrease in the $F_v/F_m$ ratio indicated that leaves were photoinhibited, a condition that molecular oxygen can represent an alternative electron acceptor for unused electrons, leading to Reactive Oxygen Species (ROS) generation [60]. Additionally, due to the inhibition in the electron transport rate, reduced activity of some $CO_2$ assimilation enzymes (carboxylase/oxygenase, ribulose-1,5-bisphosphate, and fructose-1,6-bisphosphate phosphatase) has been reported [56]. As shown in Figures 2 and 5, the observed decrease in $F_v/F_m$ ratio was coincident with a decline in leaf chlorophyll content and $CO_2$ assimilation rate. It is well-known that the lowest rate of $CO_2$ assimilation may be induced by several different factors, including water potential, limitations on $CO_2$ gas exchange, degradation of photosynthetic pigments, ion toxicity, and nutritional imbalance.

Over the experimental period, a gradual decrease in stem water potential was observed in the treated vines throughout the duration of the experiment (Figure 4). Aquea et al., 2012 [55], reported that boron toxicity induces the expression of genes involved in abscisic acid (ABA) signaling and represses genes that code for water transporters. The global changes in gene expression suggest that boron principally triggers a molecular response associated with a water-stress-related response.

In addition, significantly reduced values of $CO_2$ assimilation rate were observed in Boron-treated plots, whereas stomatal conductance was not affected. On the contrary, the period of the experimentation had a significant effect on stomatal conductance, which from day thirty to day sixty of the experimentation was reduced by a range of 30%. Additionally, the net $CO_2$ assimilation rate was reduced by 19.64 and 48.54% in control and treated vines, respectively. In addition, both the scion and rootstock cultivars had a significant effect on the net $CO_2$ assimilation rate. In general, the Merlot cultivar and 1103P rootstock showed

an increased net $CO_2$ assimilation rate, whereas the Cabernet Franc cultivar and 101-14 Mgt rootstock rated the lowest values (2.76 μmol $CO_2$ m$^{-2}$s$^{-1}$). Even though it is known that excess boron inhibits photosynthesis, information on the effects of B on the photosynthetic process is still scarce [57,61]. According to certain authors, stomatal conductance did not decrease along with the decline in photosynthetic rate in plants exposed to excess B [62]. In contrast, other authors observed a reduction in stomatal conductance [37]. The structural damage of thylakoids, according to Pereira et al., 2000 [63], was one of the potential causes of the reduction in photosynthesis caused by excess B. This, in turn, altered the rate of electron transport and the CO2 photosynthetic rate, which can also be restricted by stomatal reduction. Following B toxicity, a significant decline in the $F_v/F_m$ ratio (maximum quantum yield of chlorophyll fluorescence) was observed in many species [37,61]. Our results showed that the reduction of $CO_2$ photosynthetic rate was mainly related to a downregulation of photosystem II photochemical efficiency and partly to the stomatal conductance.

In the case of photoinhibition, because of the decrease in $F_v/F_m$ ratio, molecular oxygen can serve as an alternative electron acceptor for unused electrons and light [60], which occurs in the generation of ROS. The ROS, which are by-products of various abiotic stress, lead to dysfunction of membranes and cell death. Plants have developed a powerful scavenging system composed of antioxidant molecules and antioxidant enzymes to prevent the negative effects of these reactive molecules. Among those, phenolic substances represent a large class of secondary metabolites with strong antioxidative properties. As shown in Figure 3, sixty days from the beginning of boron treatment, leaf phenolic substances were increased by boron stress. In addition, no significant differences were found between stressed scion and rootstock cultivars.

Phenolic compounds protect plants from physiological stresses, such as oxidative stress, by preventing breakdown of macromolecules and cellular membranes. It has been reported that the antioxidant role of these compounds is due to their molecular structure [64].

## 5. Conclusions

Based on the present work, it can be concluded that the treatment with excess boron (0.5 mM B) in a hydroponic culture for sixty days negatively affected vine growth and development. Excess boron changes the ion balance of plants by influencing the elements' uptake from the nutrient media. Most macro and micronutrients responded negatively to boron excess. Vine leaves showed the highest boron concentration, followed by root concentration and the concentration of the other parts. Own-rooted treated vines and those grafted onto 101-14 Mgt rootstock concentrated increased boron in leaves that previously showed the evident symptoms of leaf damage. Yellowing of leaf margins and necrosis appeared first after twenty-five days of treatment. Additionally, the above toxic B concentrations caused an alteration in photosynthetic pigments and a downregulation of PSII photochemical efficiency resulting in a decrease of the $CO_2$ assimilation rate. Furthermore, the leaf's total chlorophyll was affected by excess boron. The excess B resulted in a significant decline of the leaf total chlorophyll concentration from 35.46% to 44.45%, thirty and sixty days, respectively, after the beginning of treatments.

It has also been confirmed that phenolic compounds increased under boron toxicity. This constitutes a common reaction of plants to avoid extensive oxidative damage. Boron-stressed vines grafted onto 1103P rootstock showed increased shoot and root dry weight, stem water potential, and net $CO_2$ assimilation rate.

It is expected that the present work will contribute to better grapevine management and growing in regions with an increased risk of boron toxicity. However, more research is required to assess more rootstocks used in grape growing.

**Author Contributions:** Conceptualization, K.-E.N.; data collection, K.-E.N., S.T. and T.C.; data analysis, K.-E.N.; writing and editing, K.-E.N.; nutrient elements analysis, K.-E.N. and T.C.; review, S.K. and A.A. All authors have read and agreed to the published version of the manuscript.

**Funding:** This research received no external funding.

**Institutional Review Board Statement:** Not applicable.

**Informed Consent Statement:** Not applicable.

**Data Availability Statement:** The data presented in this study are available on request from the corresponding author. The data are not publicly available due to privacy reasons.

**Conflicts of Interest:** The authors declare no conflict of interest.

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
