# Peer review of "Physiological and Nutritional Responses of Ungrafted Merlot and Cabernet Sauvignon Vines or Grafted to 101-14 Mgt and 1103P Rootstocks Exposed to an Excess of Boron"

_horticulturae, doi:10.3390/horticulturae9040508_

Round 1

Reviewer 1 Report

The research article entitled “Physiological and nutritional responses in to own-rooted Merlot and Cabernet Sauvignon grapevine cultivars or grafted onto 101-14Mgt and 1103P rootstocks, exposed to an excess of Boron.” by Nikolaou and co-authors discussed the effect of boron nutritional stress on grapevine cultivars with special reference to physiological and nutritional characteristics. The experiment is well designed and overall the study is interesting for the researchers and agrofarmers.

 The MS is acceptable after major revision. The specific comments/ suggestions are as stated below:

-          The MS title should me change

In Abstract

-          Abstract should be improved, not well organized, for ex. “stem water potential, net CO2 assimilation rate, and PSII maximum quantum yield have significantly been reduced”, how to guess the reduction loss as compare to control

-          Another ex., “Stressed vines grafted to 1103 P appeared to have increased shoot and root dry weights, leaf chlorophyll content, CO2 assimilation rates, and Fv/Fm ratio”

-          Authors not mention any key message   

Introduction

-          Please mention the narrow range between phytotoxic and deficient level, and cite current references. Cited ref. is too old

-          Authors wrote as “According to recent works,…….” but cited references is old, plz update and incorporate current study

-          Please revise Introduction section carefully

-          Improve the objective of the present study

M & M Section

-          According to “Treatment started at the beginning of July, for sixty days”, authors monitored the climatic variables during experimentation, if yes, plz mention

-          Why authors select only 0.5 mM of B concentration, not more or higher concentration, any specific reason behind this?

-          For chlorophyll content, I think the water bath temperature is little bit higher, ‘79.8 ° C’ plz check

-          During gas exchange measurement: what is the leaf temp., PPDF, CO2 level and observation time?

Results and Discussion section

-          Please rewrite the Table 1 caption

-          2-para- Please redraft

-          Entire the result section, data presentation is not good, please rewrite

-          Please update the discussion and incorporate recent references. Authors cited old ref.

-          Conclusion section should be improved 

Reviewer 2 Report

The manuscript measured many indicators to explore the impact of B to vine plants. The workload is sufficient, but the current manuscript is rough, and can be further improved.

some suggestions:

1) The author needs to pay attention to some detailed problems, such as CO2, '2' subscript; 101-14Mgt or 101-14 Mgt; 0.5mM, should have a space between the number and the unit; and so on. The author should carefully check the full manuscrip.

2) In table 1, the abbreviation , 'M', 'CF', 'OR', should be state clearly. And, The author said:"different letters represent significant differences". But how to compare? between different roorstock or different treatments? should state clearly.

3) The figure 1 is not clear, especially figure 1b.

4) Table 4, the results of 'Cu' is colored in red, Why?

5) In Figure 3, it seems like all the SD values are same. Some error?

6) Figure 5a, the different colored bars don't have Figure legends.

7) Finally, I want to know how the different rootstocks influence the B concentration. The whole manuscript doesn't give this conclusion. maybe should be discussed or added in the conclusion.

Round 2

Reviewer 1 Report

The authors incorporated all comments/ suggestions in the revised MS. No further corrections are required. The article can be acceptable in the Horticulturae journal.

Author Response

I would like to thank you for your review. 

Spelling check has done, as well as, some changes in the text in order to reduce the repetition rate, as demanded by editors.

You can find in attached file, all the changes in the manuscript.

Reviewer 2 Report

The author has revised manuscript according to my suggestions.

Author Response

(The authors gave the same response as above.)
